# Cardiovascular Response and Locomotor Demands of Elite Basketball Referees During International Tournament: A Within- and Between-Referee Analysis

**DOI:** 10.3390/s24216900

**Published:** 2024-10-28

**Authors:** Haris Pojskić, Edin Užičanin, David Suárez-Iglesias, Alejandro Vaquera

**Affiliations:** 1Department of Sports Science, Linnaeus University, 392 31 Kalmar, Sweden; 2Faculty of Physical Education and Sports, University of Tuzla, 75000 Tuzla, Bosnia and Herzegovina; edin.uzicanin@untz.ba; 3VALFIS Research Group, Faculty of Physical Activity and Sports Sciences, Department of Physical Education and Sports, Institute of Biomedicine (IBIOMED), University of León, 24071 León, Spain; avaqj@unileon.es; 4Faculty of Physical Activity and Sports Sciences, University of León, 24071 León, Spain; 5School of Sport and Exercise Science, University of Worcester, Worcester WR2 6AJ, UK

**Keywords:** internal and external game load, heart rate, distance covered, speed zones, acceleration and deceleration, basketball, officiating, warm-up, re-warm-up

## Abstract

There is little knowledge about within- and between-referee variation (WBRV) in cardiovascular responses (CVR) and locomotor game demands (LMD). Thus, the primary aim of this study was to assess the WBRV of CVR and LMD in male basketball referees during elite international games in preparation [e.g., warm-up (WU) and re-warm-up (R-WU)] and active game phases. The secondary aim was to explore quarter-by-quarter differences in CVR and LMD. Thirty-five international male referees took part in this study (age, 40.4 ± 5.4 years; body height, 184.9 ± 5.7 cm; body weight, 85.1 ± 7.5 kg; BMI, 24.0 ± 1.7 kg × m^−2^; fat%, 18.8 ± 4.7% and VO_2max_, 50.4 ± 2.2 L × kg^−1^ × min^−1^. In total, 76 games (e.g., 228 officiating cases) were analyzed during the FIBA elite men’s competition. They officiated 4.5 games on average (range 3–9 games). Each referee used the Polar Team Pro system to measure CVR [e.g., heart rate (HR), time spent in different HR intensity categories] and LMD (e.g., distance covered, maximal and average velocity, and number of accelerations). Results showed that the referees had bigger WBRV during the active and preparation (e.g., W-U than R-WU) phase when variables of higher CVR and LMD intensity were observed (e.g., time spent at higher HR zones, distance covered in higher speed zones). The WBRV, CVR, and LMD were higher during WU than R-WU. Moreover, the referees had a lower CVR and LMD in the second half. In conclusion, the referees should establish and follow consistently a game-to-game preparation routine and attempt to spread their on-court preparation time equally within the crew. A half-time preparation routine should be improved to re-establish a sufficient activation level similar to that achieved in pre-game preparation.

## 1. Introduction

Basketball referees are increasingly the focus of academic scrutiny due to their unique profession, where both the psychological and physical demands are high [1,2]. Given that decision-making is the most important aspect of their job [3], the investigation of factors that might affect it, such as internal and external load imposed by the game, is of crucial interest. While offering key insights into physical and physiological requirements [2], this burgeoning research field faces challenges arising from diverse methodologies, sample sizes, genders, officiating statuses, game formats, competition settings, and game periods (e.g., passive and active). For instance, referees’ cardiovascular stress and physical demands during international games were significantly lower for women, youth, and games played in the playoff phase compared to men, seniors, and games played in the group phase [4].

Moreover, it is known that the inclusion of total on- and off-court game time, such as pre-game warm-up, half-time re-warm-up, and between-quarter time, increases the total distance covered, but decreases average movement intensity and cardiovascular response compared to studies where these passive periods have been excluded from the analysis. For instance, Suárez Iglesias et al. [4] conducted a comprehensive study involving 123 elite male referees across 283 international games. This research encompassed the entire game-day experience, including warm-ups and rest periods, and found referees operated at 60–65% of their maximal heart rate (HR), notably lower than levels reported in studies focusing only on active game time [5,6,7,8,9,10,11]. Similarly, Matković et al. [7] found that referees spent 50% of the entire game in high aerobic load zones, which increased to 60% when only active quarters were considered.

In addition, a particularly contentious area of investigation is the cardiovascular demand experienced during different game quarters. While Leicht [6] found no significant HR variations across game quarters, subsequent studies have indicated a decline in HR as games progress [5,9,10,11,12,13]. Adding complexity, Godoy-Hernández et al. [14] have observed fluctuating cardiovascular demands between quarters. As for external load metrics, gathered via pedometers, ultra-wideband technology, and micro-sensors, discrepancies are equally prominent. Leicht et al. [15,16] noted variations in PlayerLoad™ to be influenced by game quarter and player gender, while Rojas-Valverde et al. [17] and Godoy-Hernández et al. [14] indicated decreasing performance variables and increasing neuro-muscular loads, respectively, in later quarters.

Furthermore, given that warm-up and re-warm-up routines are known to positively impact both athletic and cognitive performance as well as injury prevention [18,19,20,21], pre-game preparation can also be considered an essential factor for successful officiating [22,23]. Considering its importance, FIBA recommends standardized preparation protocols for both warm-up and re-warm-up. However, it seems that in praxis, many referees engage in warm-ups, but only a third of 628 referees from 18 Spanish regional organizations adhere to recommended re-warm-up protocols [4]. The literature, however, has scant coverage of external and internal load imposed on referees during warm-up (e.g., on-court pre-game preparation) and re-warm-up (e.g., on-court half-time preparation) activities, which warrants investigation in this area.

Likewise, the existing literature scarcely explores the variation in imposed game load between referees officiating the same game, despite differences in their roles and experience within the three-referee format [14]. An exception is Allegretti Mercadante et al. [24], who detected varying movement patterns among officials through video tracking. They found differences between referees in covered distance and ranges of achieved velocities. Moreover, in their recent study, Ibáñez et al. [25] found low to moderate variability (e.g., within performances, between matches, and between quarters) in external and internal load when observing four referees officiating the international tournament (2.5 ± 0.5 games/referee). In the best possible scenario, the imposed game load should be evenly spread between the referees officiating the same game. However, game-related factors such as varying amounts of court rotations [3,25,26], differences in fitness levels and training habits [27], and external factors like accumulated fatigue from increased game density during tournaments [28], as well as lack of sleep or jet lag [29], likely contribute to the un-even distribution of locomotor and cardiovascular stress among referees during active game phases.

On the other hand, between-referee variability is expected to be lower during the warm-up and re-warm-up periods. At the elite level, referees consistently perform these routines, as noted by Suárez-Iglesias et al. [4], which reduces variability during these phases. This is further supported by their adherence to FIBA’s standardized guidelines, which direct referees to evenly distribute preparation time within the crew, following a rules-dependent framework [30]. For instance, one referee monitors the teams’ warm-up from the sidelines, while the other two actively warm up. In this context, the time allocated for warm-ups is often constrained by external factors, such as team presentations or national anthems before the game (e.g., during the European Basketball Championship).

The other under-researched phenomenon in the literature is a game-to-game within-referee variability in the game load. In exercise science, considering test-retest settings where testing conditions are supposed to be un-changeable, an acceptable within-subject variation is less than 10% [31,32]. However, a higher game-to-game within-referee variation can be expected for a variety of reasons that make observed conditions changeable, such as differences in game-to-game team tactics (e.g., the game flow, different teams playing), timing of game (e.g., morning, afternoon, night time), and accumulated fatigue (e.g., lack of game-to-game rest period in tournament-based settings). Conversely, in an ideal situation, within-referee variation during warm-up and re-warm-up should be very low, as an indication of a well-established individual preparation routine. However, the current literature lacks evidence-based data on both the between- and within-referee variation in game demands during the active and passive (e.g., warm-up and re-warm-up) game phases. Moreover, as previously suggested, a systematic study that would include a larger number of referees was warranted [25].

Given these gaps in the existing literature, our primary research aim was to assess within- and between-referee variability of the cardiovascular responses and locomotor demands in male basketball referees during elite international games, both in active and preparation phases. Our secondary aim was to explore quarter-by-quarter differences in these physiological variables to inform effective warm-up strategies and targeted training following quarter-specific demands. We hypothesized that: (i) there would be significant variability in cardiovascular and locomotor loads both between referees officiating the same games and within individual referees across consecutive games in the tournament; and (ii) physiological demands would vary by quarter, with higher loads expected in the first half and during the initial warm-up compared to the second half and re-warm-up periods.

## 2. Materials and Methods

### 2.1. Study Design

This investigation was an observational cross-sectional study of referees’ cardiovascular and locomotor responses during the FIBA Men’s EuroBasket, which took place from 31 August to 17 September 2017. In total, 76 games (228 officiating cases) were analyzed. Each referee officiated at least three games (range: 3–9 games). See Figure 1 (the studies’ phases).

### 2.2. Participants

Thirty-five internationally licensed male referees from 25 countries volunteered and took part in this study. Their anthropometric, physiological, and demographical information is presented in Table 1. All referees had completed a general health prescreening questionnaire and were classified as healthy (i.e., no known disease) before the beginning of the tournament. All referees undertook the same 12-week training program prescribed by the FIBA referees’ fitness team [11], consisting of three weekly sessions combining endurance training (e.g., 60–70% HR_max_ intervals or Fartlek at 60–80% HR_max_), speed-focused repeated sprint ability drills (e.g., 80–90% HR_max_ with incomplete recovery), strength training, and game practice. Each session followed a structured format of warm-up (e.g., 5–10 min jogging at 50–60% HR_max_), main training exercises, and recovery (foam rolling). Training intensity was monitored via HR zones, progressively increasing in volume and intensity throughout the program. None of them reported taking any medications that could affect HR. All referees were informed of the purpose, benefits, and risks of the study before providing written informed consent to participate. All investigative procedures were conducted in accordance with the World Medical Association International Code of Medical Ethics and approved by an institutional ethics committee.

### 2.3. Procedures

FIBA arranged the tournament schedule and allocated referees through its Referee Department, considering factors such as referee experience, past performances, teams playing, and necessary rest intervals. Referees officiated daily, with at least 16 h of rest between games. Each game was managed by three referees, adhering to FIBA official rules. The group phase was played in four different countries, with each group playing all its games in a single country. The round of 16, quarterfinals, semi-finals, the 3rd place game, and the final game were all played in the same arena in one of the countries from the group phase. The daily schedule varied significantly across the tournament, with the first game starting at 12.30 PM and the last game at 9.30 PM.

### 2.4. Pre-Game Preparation

The FIBA Referee Department provided standardized protocols for warm-up and recovery, both on-court and off-court. Off-court warm-up routines, conducted in the locker room, included self-myofascial release (e.g., foam rolling), static and dynamic stretching, and muscle activation exercises (e.g., squats, double- or single-leg bridges). The on-court warm-up was performed in rotation, allowing one referee to observe the court while the others warmed up on the sidelines. It consisted of low-intensity running and dynamic stretching, followed by several minutes of higher-intensity exercises, such as short sprints and change-of-direction drills. During the 15-min halftime break, referees were asked to passively recover in the locker room for 5–7 min and then to perform a re-warm-up on the court following the recommendations given for the warm-up. Post-game routines included a meeting at the scorer’s table for approximately 5 min, followed by a 5-min passive static stretching and cold application (e.g., ice bags) in the locker room.

To measure indoor velocity, distance, and HR, the Polar Team Pro System (Polar Electro OY, Kempele, Finland) was used. This system integrates various sensors (e.g., 10 Hz GPS, accelerometer, gyroscope, digital compass, and sampling at 200 Hz) and includes in-built HR monitoring with proprietary software. Non-GPS sensors and proprietary algorithms were used to calculate velocity and distance, making this system suitable for indoor settings, and enabling efficient analysis of external workload data [33]. This micro-sensor system has been used in indoor sports such as futsal, basketball, and handball [34,35,36]. The Polar Team Pro system has shown reliability in measuring HR responses [37,38] and locomotor activities (e.g., velocity and distance) in outdoor environments [39].

Following manufacturer guidelines, each referee wore the sensor attached to an elastic strap around the lower sternum. To reduce inter-device variability, the same sensor was used by each referee [34]. The sensor was activated 20 min before the game and worn until 5 min post-game, recording passive and active periods during pre-game and post-game times, as well as actual playing time with applicable stoppages (e.g., free throws, timeouts, fouls, and violation calls) [40]. The recording session was uploaded to a local computer using the manufacturer’s interface and online solution (Polar Team Pro System) for subsequent analysis, which included separate quarter-by-quarter and warm-up and re-warm-up investigations for each referee officiating the same game (Figure 2).

### 2.5. Measurements

#### 2.5.1. Anthropometrics 

The FIBA Referees’ Fitness Coordinator, who holds a PhD in Physical Activity and Sport Sciences, collected all anthropometric data using the same instruments and under uniform environmental conditions. These measurements were taken the day before the tournament started, during a single session held from 08:00 to 08:30 AM, following an overnight fast [12]. Body mass and height were measured with a digital scale (Seca Alpha, GmbH & Company, Igni, France; range 0.1–150 kg, accuracy 0.01 kg) and a Harpenden digital stadiometer (Pfifter, Carlstadt, NJ, USA; range 70–205 cm, accuracy 1 mm), with referees dressed only in their underwear. Body fat percentage was determined using electrical bioimpedance (Tanita OMRON BF306, Arlington Heights, IL, USA). Fat-free mass was calculated using the formula [(FFM = body mass − (body mass × bioimpedance body fat percentage)] [41].

#### 2.5.2. Aerobic Capacities (VO_2max_ Measurements)

During the FIBA Men’s EuroBasket Pre-Competition Clinic (PCC), held in Istanbul from 25 to 30 August 2017, referees completed the mandatory FIBA aerobic fitness test (e.g., the multi-stage shuttle run test) on 27 August. The PCC was a six-day intensive preparation program aimed at standardizing the physical, mental, and technical readiness of all referees. For the two days leading up to the test, referees were housed in the same hotel and followed identical routines, including structured mealtimes and rest periods, ensuring uniform conditions and preparation across the group.

During the test, participants ran 20-m shuttles back and forth at a set pace dictated by pre-recorded beep signals. They started at a speed of 8 km·h^−1^, increasing their speed by 0.5 km·h^−1^ every minute. The last completed shuttle was used to estimate their VO_2max_ [42]. The test reliability was shown to be excellent (ICC = 0.95) for adult men and women [43].

#### 2.5.3. Cardiovascular and Locomotor Responses During Games 

HR responses were recorded in absolute terms (beats·min^−1^) and normalized to each referee’s HR_max_ achieved during the mandatory FIBA aerobic fitness test to reflect relative exercise intensity [10,11]. HR data were exported and analyzed in Microsoft Excel (v19.0; Microsoft Corporation, Redmond, WA, USA) to determine the percentage of time spent in different HR intensity categories [11]. The HR categories were: very hard, >89% HR_max_; hard, 80–89% HR_max_; moderate, 70–79% HR_max_; light, 60–69% HR_max_; and very light, 50–59% HR_max_ [44]. For locomotor activities, average velocity, total distance, and mean distance covered by referees were calculated, along with distances covered within specific velocity categories: very hard, ≥19 km·h^−1^; hard, 15–18.99 km·h^−1^; moderate, 11–14.99 km·h^−1^; light, 7–10.99 km·h^−1^; and very light, 3–6.99 km·h^−1^. The number of changes in direction (e.g., decelerations and accelerations) was estimated within different intensity categories based on the following zones: Z1 = 0.5–1.0 m·s^−2^ (low intensity); Z2 = 1.0–2.0 m·s^−2^ (moderate intensity); Z3 = 2.0–3.0 m·s^−2^ (high intensity); Z4 ≥ 3.0 m·s^−2^ (very high intensity) [45].

#### 2.5.4. Statistical Analysis

Cardiovascular and locomotor response data were extracted and presented separately for warm-up (20 min), each quarter, re-warm-up (15 min), and total active time (sum of all quarters). Data are presented as means and standard deviations (SDs). The Shapiro–Wilk test was chosen to assess normality, as it is suitable for smaller sample sizes, ensuring the appropriateness of subsequent parametric tests. The within- and between-referee variation for all outcome variables was established using the coefficient of variation (CV%) expressed as a percentage [31], as it provides a standardized measure of dispersion relative to the mean, making it ideal for comparing variability across referees and game conditions. Within-referee variation (CV%), mean, and SD were derived based on the following calculations (R_G1_ + R_G2_ + R_G3_ + … + R_Gn_)/n, where R was a referee, G was an officiated game, and n was a number of officiated games during the tournament. Between-referee variation (CV%), mean and SD were calculated based on the following calculation (R_1_ + R_2_ + R_3_)/3, where the numbers 1, 2, and 3 represented three different referees who officiated the same game. The following criteria were used to interpret CV%: low (<10%), moderate (10–30%), and high (>30%) [46]. Repeated measures ANOVA was used to identify the within-subject differences in cardiovascular response and locomotor demands across the quarters. Bonferroni post-hoc tests were used for pairwise comparisons (quarters: 1st, 2nd, 3rd, and 4th). The repeated measures ANOVA allowed us to control for individual variability, focusing on how referees’ cardiovascular and locomotor responses changed over time within the same individual. Partial eta squared (η_p_^2^) was calculated for the ANOVA main effects, with effect sizes (ES) interpreted as follows: >0.02 (small), >0.13 (medium), and >0.26 (large) [47]. Statistical significance for all tests was set at *p* ≤ 0.05. Statistical analyses were performed using SPSS^®^ 25.0 (IBM, New York, NY, USA).

## 3. Results

The Shapiro–Wilk test showed that data for all measures were normally distributed. Descriptive statistics (mean ± SD and CV%) were calculated for all outcome variables, including age, anthropometrics, aerobic capacity, and history of officiating experience. The biggest between-referee variation was in years of international officiating experience (11.31 ± 5.92 years) and number of officiated games per season (61.05 ± 20.02). At the same time, there was no substantial variation in anthropometric measures and VO_2max_ (Table 1).

A total of 76 games, including the group and playoff phases, were analyzed, producing 228 individual datasets. Each referee officiated in an average of 4.5 games (range: 3–9) with an average duration of active officiating time of 85.2 ± 8.9 min per game. The duration of the 1st quarter (18.0 ± 2.2 min) was significantly shorter than that of the 2nd, 3rd, and 4th quarters (23.1 ± 4.1, 20.2 ± 3.0, 24.0 ± 5.2 min, respectively), and the 3rd quarter was significantly shorter than the 2nd and 4th quarters (Table 2).

The average game HR was 126.3 ± 23.5 beats·min^−1^, which was related to light exercise intensity (68.1 ± 12.9% HR_max_), while the average HR_max_ was 82.2 ± 15.3%. Referees on average spent the most time in HR zones Z2 and Z3 (25.6 ± 10.9 and 28.5 ± 10.0 min, respectively). They covered in average 3034.8 ± 422.4 m per game within different speed zones (Z1 = 1402.0 ± 233.0 m, Z2 = 627.40 ± 96.2 m, Z3 = 443.9 ± 96.9 m, Z4 = 255.2 ± 70.7 m, Z5 = 94.0 ± 37.0 m). The average maximum speed across the tournament was 15.3 ± 3.2 h^−1^, with the highest achieved being 21.0 ± 2.4 h^−1^. They had an average of 1105.3 ± 147.6 changes in direction per game, with most accelerations in Z1 and Z2 (197.1 ± 38.8 and 304.4 ± 43.9, respectively) and most decelerations in Z3 and Z4 (307.7 ± 42.2 and 163.2 ± 27.0, respectively).

### 3.1. Differences in Cardiovascular Responses Across the Quarters

HR_avg_ (beats·min^−1^) and HR_avg_ (%) significantly differed between the 2nd quarter (132.6 ± 9.9 and 71.4 ± 5.8%) and the 3rd (128.4 ± 9.5 and 69.2 ± 5.7%) and 4th quarters (127.0 ± 10.1 and 68.4 ± 6.0), with no further significant differences. The HR_max_ (beats·min^−1^) and HR_max_ (%) were significantly different between the 2nd (160.5 ± 9.5 and 86.5 ± 5.7) and the 3rd (154.7 ± 11.0 and 83.2 ± 6.6) and 4th quarter (154.6 ± 11.4 and 83.2 ± 6.7), while there were not any other differences.

The referees spent more minutes in HR zones Z4 and Z5 during the 1st (4.0 ± 2.8 and 0.8 ± 1.03) and 2nd quarters (4.2 ± 3.46 and 0.7 ± 1.0) than the 3rd (2.6 ± 2.2 and 0.3 ± 0.4) and 4th quarters (2.9 ± 2.8 and 0.4 ± 0.5). Conversely, they spent fewer minutes in HR zones Z1, Z2, and Z3 during the 3rd (2.5 ± 1.8, 6.9 ± 3.3, and 7.0 ± 3.2) and 4th quarters (3.2 ± 2.5, 8.5 ± 4.4, and 7.8 ± 3.6) compared to the 1st (1.5 ± 1.4, 4.2 ± 2.6, and 6.2 ± 2.6) and 2nd quarters (2.1 ± 2.1, 6.8 ± 4.1, and 8.6 ± 3.8).

### 3.2. Differences in Locomotor Demands Across Quarters

Referees covered more distance in the 2nd (842.5 ± 179.3 m) and 4th (813.8 ± 192.9 m) quarters compared to the 1st (708.0 ± 147.0 m) and 3rd (757.2 ± 127.2 m) quarters. In the 1st and 2nd quarters, they accumulated more distance in speed zones 3, 4, and 5 than in the 3rd and 4th quarters. In contrast, in the 4th quarter, they covered the most distance in the slowest speed zone (Z1). The majority of changes in direction speed occurred in the 2nd (305.8 ± 45.4) and 4th (302.8 ± 45.3) quarters. They performed a higher number of high-intensity accelerations and decelerations in the first two quarters than in the last two quarters.

### 3.3. Within-Referee Variability in Cardiovascular Responses and Locomotor Demands During Preparation Time

Within-referee variation (CV%) in cardiovascular response was greater during warm-up than re-warm-up. For HR_avg_ (beats·min^−1^), HR_max_ (beats·min^−1^), HR_avg_ (%), and HR_max_ (%), the values during warm-up were 16.2, 15.0, 16.6, and 15.6%, respectively. In comparison, the values were 8.7, 10.4, 9.4, and 10.8% during re-warm-up. The CV% dramatically increased for time spent in different HR zones from Z1 to Z5 during both the warm-up (63.4–185.3%) and re-warm-up (59.1–197.1%) periods. The cardiovascular response was higher in the warm-up compared to the re-warm-up period.

Within-referee variation in distance covered was slightly lower during warm-up (30.7%) compared to re-warm-up (32.7%). The difference was more prominent in terms of variation in maximum speed (km·h^−1^) achieved during warm-up (26.0%) compared to re-warm-up (39.8%). The CV% notably increased for distance covered across different speed zones from Z1 to Z5, both during warm-up (33.0–89.2%) and re-warm-up (21.5–162.4%). A similar pattern was noted concerning variation in changes in direction speed. The higher the intensity of acceleration or deceleration, the greater the within-referee variation. The total number of changes in direction speed varied from 29.0% during warm-up to 28.9% during re-warm-up. Maximum speed was higher during the warm-up (18.7 ± 4.3 km·h^−1^) compared to the re-warm-up (13.3 ± 5.1 km·h^−1^).

### 3.4. Within-Referee Variability in Cardiovascular Responses and Locomotor Demands During Active Time

Within-referees variation in cardiovascular response averaged 10% when analyzed separately for each quarter and approximately 19% for the entire active time. There was a clear tendency for increased variation (approximately 40–160%) regarding time spent in different HR zones, from lower to higher intensity, for both total active time and individual quarters.

The variation in distance covered differed across quarters and total distance, with values of 22.8%, 21.7%, 17.3%, 23.9%, and 15.5%, respectively. Maximum and average speeds varied across the quarters and overall, ranging from 11.7% to 21.2%. An evident trend was observed for increasing variation (approximately 20–90%) in distance covered across different speed zones as speed intensity increased. A similar pattern was noted for variations in changes of direction speed. The higher the intensity, the greater the variation (approximately 20–183%).

### 3.5. Between-Referee Variability in Cardiovascular Responses and Locomotor Demands During Preparation Time

The between-referee variation was higher during warm-up compared to re-warm-up in HR_avg_ (27.5 and 8.1%), HR_max_ (26.6 and 7.0%), HR_avg%_ (28.1 and 9.1%), and HR_max%_ (26.5 and 7.6%) (Table 3). In general, the higher the HR zone, the greater the variation observed. The highest variation was in Z4 and Z5 during both the warm-up (100.5% and 159.5%) and the re-warm-up (123.9% and 139.7%). There was also greater variation in calories burned during the warm-up (34.4%) compared to the re-warm-up (22.2%).

The variation in distance covered was greater during warm-up (48.3%) compared to re-warm-up (16.3%). Maximum and average speeds also fluctuated more during warm-up (42.1% and 48.3%) than during re-warm-up (27.4% and 17.1%). The largest variation was found in the distance covered within speed zones Z3, Z4, and Z5, during both warm-up (77.7%, 106.5%, and 140.8%) and re-warm-up (104.2%, 118.7%, and 143.8%). Generally, the more intense the speed zone, the higher the observed variation. Referees showed a larger variation in the total number of changes in direction speed during re-warm-up (52.9%) compared to warm-up (12.5%). Considerable variability was also noted in high-intensity acceleration and deceleration zones during both warm-up (165.3% and 135.6%) and re-warm-up (163.2% and 159.0%).

### 3.6. Between-Referees Variability in Cardiovascular Responses and Locomotor Demands During Active Time

The variation in HR_avg_ and HR_max_ for the entire game ranged between 10 and 12%. The greatest variability was noted in the time spent in different HR zones, with the biggest variation identified in the highest intensity zone Z5 (approximately 150%) and the lowest in Z1 (approximately 103%). The most pronounced variation in HR_avg_ and HR_max_ occurred during the 1st quarter (approximately 24%), whereas it ranged between 7 and 12% in the other quarters. Regarding the time spent in different HR zones, the highest variation was in zones 3 and 4 across all quarters, ranging from approximately 82 to 167%. The greatest variation in energy expenditure was detected in the 2nd quarter (106%).

The average variation during total playing time in distance covered, maximum, and average speed was 14.8%, 17.4%, and 14.6%. The most substantial variation for these variables was found in the 1st quarter (28.3%, 31.3%, and 28.3%, respectively). Referees showed the most variability in distance covered in the highest speed intensity zone, Z5, across all quarters (approximately 52–100%). The variation in the overall number of changes in direction speed during a game was approximately 23%. The greatest between-referee variation was observed in high-intensity acceleration and deceleration zones, ranging from 80 to 180%.

## 4. Discussion

To the best of our knowledge, this is the first study to investigate within- and between-referee variability of the cardiovascular responses and locomotor demands of male referees during elite international basketball games, both in game and preparation phases (e.g., warm-up and re-warm-up) of the game. Several important findings from this study should be acknowledged. First, the referees demonstrated greater between- and within-referee variation during both active and preparation game time when higher cardiovascular and locomotor intensity variables were measured. Second, both within- and between-referee variations were higher during the warm-up than the re-warm-up. Third, the referees exhibited lower cardiovascular response and locomotor demands in the 2nd half. Fourth, cardiovascular response and locomotor demands were higher during the warm-up than during the re-warm-up period.

Across the observed tournament, referees experienced an average HR during active periods of 68.1 ± 12.9% of HR_max_ with most time (approximately 63%) spent in HR zones Z2 and Z3, which corresponded to moderate to light exercise intensity. The cardiovascular response was expectedly higher compared to the results in the recently published study by Suárez-Iglesias et al. [4]. In brief, they reported an average relative HR of 60–65% of maximum HR in 123 elite male referees while officiating 283 international basketball games. Their analysis included warm-up, re-warm-up, and quarter breaks. The mean session time was approximately 120 min, compared to 85.2 ± 8.9 min in the current study. Consequently, the HR response in the current study was higher than in their study, but still lower (i.e., range between 7 and 22%) than that reported for international and national competitions that only recorded responses during game time without any breaks [5,6,7,8,9,10,12,40].

Moreover, the average covered distance (3.06 ± 0.42 km) was shorter (approximately 1–2 km) compared to previously reported distances (4.82 ± 0.67 km, 4.02 ± 0.62 km, and 4.52 ± 0.49 km) [4,5,24,25]. Observed discrepancies could be attributed to the different measurement methodologies employed. The current study used an accelerometer, whereas others used video time-motion analysis [8,24], pedometers [5], or positioning systems [15,16,25]. Additionally, as mentioned earlier, our research focused exclusively on active playing time, excluding warm-up and re-warm-up periods, while other studies incorporated recordings commencing 20 min before the beginning of a game, encompassing both passive and active intervals [4,40]. Consequently, the chosen measurement methodology can result in approximately 1–2 km differences in distance covered per game.

In addition, referees covered most distances (approximately 2.5 km) in lower speed zones (Z1-Z3), which was lower than previously published values. However, it is important to note that referees covered more than 350 m on average in the two highest speed zones, Z4 and Z5 (15.00–18.99 km·h^−1^ and ≥19.00 km·h^−1^), which aligns with previously published data [4,25]. This high-speed coverage, together with the total average accelerations and decelerations per game (1105.3 ± 147.6), might impose significant muscular load accompanied by moderate-to-high cardiovascular response.

### 4.1. Differences in Cardiovascular Responses Across the Quarters

Interestingly, only a few studies have explored quarter-by-quarter differences in cardiovascular and locomotor load. Notably, reduced cardiovascular response and locomotor demands were observed in the 2nd half. This finding contrasts with Leicht [6], who found no significant heart rate variations across game quarters. Conversely, our findings corroborate several studies indicating a decline in cardiovascular response [5,9,10,11,13], performance variables [14,17,25], and training load [15,16,25] as games progress. For instance, Ibáñez et al. [25], in the recent study reported less distance covered, but higher neuro-muscular load in the first quarter than in the rest of the games. This load was induced by a greater number of high-intensity activities (e.g., sprints, accelerations, and decelerations). In the current study, referees spent more minutes in high-intensity HR zones during the 1st and 2nd quarters than in the 3rd and 4th quarters. This increase in high-intensity effort during the 1st half was followed by reduced locomotor demands in the 2nd half (e.g., fewer high-intensity activities and less distance covered in high-intensity speed zones). A lower overall load during the 2nd half could be explained by a longer 2nd half (approximately 44 min) than the 1st half (approximately 31 min), with the 4th quarter being the longest (24.0 ± 5.2 min). This is likely due to several factors. First, teams make more conservative and careful strategic decisions that slow the pace towards the end of the game [48]. Specifically, teams tend to have longer ball possessions by reducing the number of risky passes and turnovers, which decreases opponents’ opportunities for fast breaks and easy scores [48,49]. Second, there is a longer passive game time because of the increased number of “tactical” fouls and subsequent free throws, as well as a higher number of timeouts and substitutions, especially evident in close, balanced games. Collectively, based on our results and prior research, a decline in cardiovascular response and locomotor demands in the 2nd half is expected. This reduction is likely driven by decreasing physical demands and cumulative mental fatigue, both of which contribute to the observed reductions in HR_avg_ and HR_max_ as the game progresses [5,9,10,11,12,13,25].

### 4.2. Within-Referee Variability in Cardiovascular Responses and Locomotor Demands

Overall, cardiovascular response and locomotor demands were greater during the warm-up compared to the re-warm-up period, which can be partially attributed to the difference in duration (e.g., 20 vs. 15 min). For instance, HR_max_ (%), distance covered, and max speed were higher during the warm-up (81.6 ± 10.9%, 515.3 ± 133.0 m, and 18.7 ± 4.3 km·h^−1^) compared to the re-warm-up (77.2 ± 8.4%, 243.4 ± 84.0 m, and 13.3 ± 5.1 km·h^−1^). It is worth noting that the achieved activation level was significantly lower in the re-warm-up than in the 3rd quarter. Specifically, the achieved HR_max_ was approximately 7–10% lower, and the maximum speed reached was approximately 7–8 km·h^−1^ slower in the re-warm-up than in the 3rd quarter. Moreover, referees performed fewer than one high-intensity movement (e.g., linear sprints) during the re-warm-up compared to 4–5 high-intensity activities in the 3rd quarter. These differences were significantly smaller when comparing the warm-up and the 1st quarter, indicating a more effective warm-up routine than re-warm-up. However, the variability in the distance covered exceeded 30% in both the warm-up and re-warm-up, potentially leading to increased variability in other metrics.

Generally, higher within-referee variability in cardiovascular responses and locomotor demands was noted during the warm-up compared to the re-warm-up period. When examining the variation in HR_avg_ (%) and HR_max_ (%), it was approximately 15% (e.g., moderate) in the warm-up and approximately 10% (e.g., low) in the re-warm-up. Consequently, it can be concluded that referees approached the warm-up and re-warm-up differently. Interestingly, a significant within-referee variation was noted both in the warm-up (63.4–85.3%) and re-warm-up (59.1–197.1%) when higher intensity cardiovascular and locomotor responses were recorded (e.g., time spent in HR Z5 and distance covered in speed zones Z4 and Z5). This suggests that referees do not perform either the warm-up or re-warm-up consistently across games, especially considering high-intensity activities (e.g., power steps, sprints), which might increase their risk of injury.

Even though there is a higher prevalence of overuse compared to acute lower-limb injuries in basketball referees [23], the importance of warm-up routines should be emphasized. Little is known about the mechanisms and timing of injuries in basketball referees, but it is known that approximately 40% of injuries in basketball players are noncontact injuries (i.e., occur after acceleration or deceleration activities), and most of them happen in the first 10 min of active playing time [50]. Moreover, it is generally understood that the implementation of warm-up and preventive strategies can significantly reduce the risk of noncontact lower-body injuries in basketball players [51,52]. Therefore, it is reasonable to conclude that the consistent implementation of a game-to-game warm-up routine could not only have athletic and mental performance benefits but also reduce injury risk in referees [18,19,20,21,23]. Based on the findings of the current study, it is recommended that referees establish and perform a consistent warm-up routine, aiming to increase the re-warm-up time on the court and to achieve a similar activation level as in the warm-up. This way, they can be adequately prepared for the beginning of the 2nd half.

Likewise, the observed preparation periods, the within-referee variability in cardiovascular response, and locomotor demands during the active time were greater when higher-intensity variables were considered. This might be due to several factors that contribute to the changeable and unique nature of the game-to-game environment, which is reflected in the variation in total game duration (e.g., 10.5%) and quarters duration (e.g., 1st = 12.0%, 2nd = 17.9%, 3rd = 14.7%, and 4th = 21.5%). First, the game rhythm and pace depend on various factors such as the number of ball possessions, applied team tactics (e.g., man-to-man or zone defense, full-court press defense), the balance of the game and teams, and how close the score is in the final, critical part of the game, which usually affects the number of tactical breaks such as timeouts, substitutions, and committed fouls [48,53,54,55].

Second, daily biological variability (i.e., diurnal variation) and different chronotypes (i.e., “early” and “late”) are suggested to influence cognitive and athletic performance [56]. Given that the daily game schedule varied considerably across the tournament, with the first game starting at 12:30 PM and the last at 9:30 PM, it can be speculated that this could increase referees’ variability in their cardiovascular responses and locomotor demands. Moreover, due to the various nomination criteria, some referees officiated the late game and the early game the following day, which could affect their quantity and quality of sleep, thereby contributing to game-to-game variability. Additionally, variability could be heightened by jet lag and travel fatigue. Specifically, the tournament took place in different time zones, and some referees had to travel across several time zones to officiate. It is known that jet lag can negatively influence diurnal rhythms in athletes, which in turn can disrupt sleep, cause daily complaints (e.g., gastrointestinal issues), impair cognitive functions (e.g., attention, complex mental tasks), reduce motivation and mood, and diminish athletic performance [57]. All of these factors can collectively impact cardiovascular response and locomotor demands. However, it is important to note that referees participated in FIBA’s 7-day pre-competition clinic, which provided them with the opportunity to adjust to the new time zone.

### 4.3. Between-Referee Variability in Cardiovascular Responses and Locomotor Demands

Generally, there was substantial between-referee variation in cardiovascular responses and locomotor demands during both warm-up and re-warm-up. Additionally, this variation was more pronounced during warm-up compared to re-warm-up, especially when examining higher-intensity variables. The percentage differences in average and maximal HR exceeded 25%, and the time spent in high-intensity HR, speed, acceleration, and deceleration zones fluctuated by more than 100%. One contributing factor could be the greater disparity in the distance covered during warm-up (i.e., 48.3%) compared to re-warm-up (i.e., 16.3%). These results suggest that the crewmembers do not perform either warm-up or re-warm-up consistently, indicating that they do not achieve the same activation level before the start of the game. It seems that they do not evenly allocate time to the recommended warm-up activities (e.g., dynamic stretching, sprints). However, ensuring equal preparation time within the crew is challenging due to basketball rules that require one of the crewmembers to stay at the sideline opposite the score table and observe the teams’ warm-up. Additionally, the pre-game preparation is interrupted several times (e.g., team introductions and national anthems), making it difficult to distribute the available warm-up time fairly among the crew. Moreover, there was greater variability in cardiovascular response during the preparation periods compared to game time, highlighting the need for better allocation of the available time within the crew and consistent pre-game activation. Similarly, during active game time, between-referee differences are evident in all measurements, with more pronounced variability in high-intensity variables. Interestingly, the highest variability is seen in the 1st quarter, which can be explained by a higher number of high-intensity activities. Interestingly, even though Ibáñez et al. [25] reported also a greater number of high-intensity activities in the 1st quarter, in general, they found low to moderate variability (less than 30%) within performances and between matches in observed external and internal load variables. This might be explained by discrepancies in the used methodology. In brief, they used an 8-antena positioning system suitable for indoor movement studies, and they analyzed a lower number of elite referees (n = 4) and officiated games (range: 2–3 games), which all together could contribute to the reported lower variability.

The between-referee variability can be partially attributed to the un-equal rotations of the referees on the court in the three-referees officiating system. In particular, if one or two referees spend more time as the trial referee during the game, it may result in more distance covered and longer sprints compared to the lead or center referees. Additionally, discrepancies in fitness level (i.e., lower fatigue resistance) might contribute to this variability, even though all referees followed the 12-week pre-competition training program [11] and passed the fitness test to be nominated for the tournament. Furthermore, factors mentioned earlier, such as jet lag, travel fatigue, diurnal rhythms, and chronotypes, could impact cardiovascular responses and locomotor demands, increasing the between-referee variability.

### 4.4. Limitations

Several limitations should be acknowledged. First, the cross-sectional research design limits the strength of the conclusions. Second, the un-equal number of games officiated by each referee, ranging from 3 to 9 games with an average of 4.5, introduces variability in game-related exposure. While we used the coefficient of variation (CV%) to account for this variability by providing a standardized measure of dispersion relative to the mean, this approach cannot fully mitigate the potential impact of factors such as cumulative fatigue or better adaptation to game demands in referees with more games. Referees who officiated fewer games may still show greater variability in their cardiovascular and locomotor responses due to the smaller sample size of games, which could affect the consistency of performance estimates and contribute to the overall variability observed in the study’s findings. Third, the analysis included all passive game periods (e.g., timeouts, instant replay reviews), which could reduce the overall response intensity. Fourth, the study did not consider potential confounding variables such as the tournament phase, differences in fitness levels among referees, game-to-game recovery time, game rhythm, and the balance of the games. Fifth, the study included only male referees, which might limit the extrapolation of the findings to female referees.

### 4.5. Practical Applications

To reduce injury risks and optimally enhance physical and mental activation, referees should adopt and consistently follow a game-to-game preparation routine. They should also aim to distribute their on-court warm-up time evenly within the crew, while respecting individual differences and preferences. Despite anticipating a lower cardiovascular response and locomotor demands in the 2nd half compared to the 1st half, we strongly recommend that referees increase the volume and intensity of their re-warm-up to achieve activation levels similar to those reached during the warm-up.

Additionally, based on the observed variability in cardiovascular and locomotor loads, referees should regularly monitor their individual heart rate and locomotor data during tournaments to adjust their preparation, ensuring consistent performance. Implementing this monitoring could guide referees in customizing their re-warm-up routines to game-specific demands. This individualized approach would help maintain optimal activation levels throughout the tournament.

Given the lack of available on-court preparation time during international tournaments, referees should maximize their pre-game off-court preparation time, including warm-ups in the locker room and, if available, in appropriate spaces outside (e.g., the corridor). Additionally, morning activation (e.g., light-intensity jogging, stretching) and “power naps” in case of sleep deprivation are advised as pre-game strategies. These practices provide day-long positive effects on cognitive functions (e.g., increased arousal, concentration, and attention) and physical performance (e.g., speed and endurance) while reducing perceived fatigue [58,59,60,61]. Despite being aware of the multiple factors that affect the nomination process, we strongly encourage the FIBA officiating department to nominate referees in a way that allows sufficient recovery time between games (e.g., at least 24 h) and to consider different chronotypes when possible (i.e., assigning “early birds” to early games and “night owls” to late games). Furthermore, we strongly support the department’s current practice of organizing pre-competition clinics lasting at least 5–7 days, providing referees from different time zones sufficient time to adapt and mitigate the negative effects of jet lag [57].

## 5. Conclusions

Moderate cardiovascular and locomotor loads were observed in the elite group of international male basketball referees, which aligns with previous studies. However, different loads were imposed on the crewmembers officiating the same games. Additionally, the same referee consistently experienced varying loads in consecutive games during the tournament. These differences were more pronounced when high-intensity variables were analyzed, both in the preparation phases and during the active periods of the game. Variability and imposed game load were higher during the 1st half and in the warm-up compared to the re-warm-up. The findings confirmed the study’s hypotheses.

## Figures and Tables

**Figure 1 sensors-24-06900-f001:**
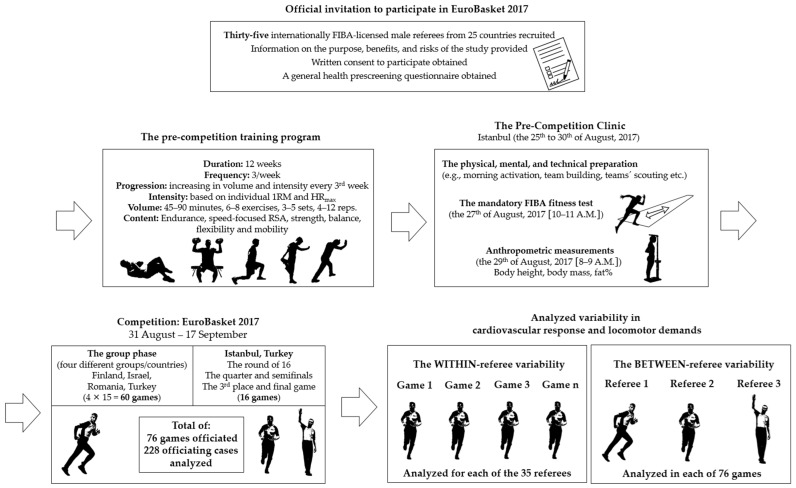
The study’s phases.

**Figure 2 sensors-24-06900-f002:**
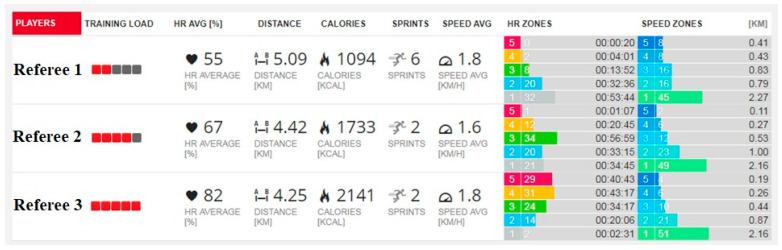
Example of comparative analysis of an officiating crew in cardiovascular responses and locomotor game demands (summarized data of active and preparation game time).

**Table 1 sensors-24-06900-t001:** Anthropometric, physiological, and demographic data of elite male international basketball referees (n = 35).

Variables	Mean ± SD	CV%
Age (years)	40.44 ± 5.41	13.39
Officiating experience (years)	22.02 ± 6.15	27.93
International experience (years)	11.31 ± 5.92	52.38
Games per season (number)	61.05 ± 20.02	32.80
Games per week (number)	2.28 ± 0.73	31.96
Body height (cm)	184.94 ± 5.71	3.08
Body weight (kg)	85.02 ± 6.37	7.50
BMI (kg/m^2^)	24.83 ± 1.16	4.68
Body fat (%)	18.77 ± 2.86	15.25
VO_2max_ (mL/kg/min)	50.44 ± 2.22	4.40

Legend: BMI = body mass index; SD = standard deviation; CV% = coefficient of variation.

**Table 2 sensors-24-06900-t002:** Descriptive statistics (mean ± SD) and within-referees variability of external and internal load data of FIBA referees (N = 35).

Variables	Warm-Up		1st Quarter		2nd Quarter		Half Time		3rd Quarter		4th Quarter		Total (Active Time)		Q by Q Comparison	
	Mean ± SD	CV%	Mean ± SD	CV%	Mean ± SD	CV%	Mean ± SD	CV%	Mean ± SD	CV%	Mean ± SD	CV%	Mean ± SD	CV%	*p*-Value	(η_p_^2^)
Duration (min.)	20 ± 0	-	18.0 ± 2.2 *^†¥^	12.0	23.1 ± 4.1 ^†^	17.9	15 ± 0	-	20.2 ± 3.0 ^¥^	14.7	24.0 ± 5.2	21.5	85.2 ± 8.9	10.5	0.01	0.34
HR_min_ (beats/min.)	80.3 ± 7.9	9.7	99.8 ± 11.3 *	11.4	105.5 ± 10.9 ^†¥^	10.5	88.8 ± 8.2	9.6	100.1 ± 7.4	7.4	101.5 ± 8.8	9.0	98.9 ± 18.9	19.2	0.00	0.40
HR_avg_ (beats/min.)	104.0 ± 15.2	16.2	131.6 ± 14.8	12.7	132.6 ± 9.9 ^†¥^	7.5	108.0 ± 9.6	8.7	128.4 ± 9.5	8.0	127.0 ± 10.1	8.5	126.3 ± 23.5	18.6	0.00	0.38
HR_max_ (beats/min.)	151.6 ± 19.6	15.0	158.1 ± 15.7	11.3	160.5 ± 9.5 ^†¥^	5.9	143.3 ± 15.0	10.4	154.7 ± 11.0	7.6	154.6 ± 11.4	7.8	152.6 ± 28.0	18.4	0.00	0.33
HR_min_ (%)	43.4 ± 4.6	10.3	53.9 ± 6.6 *	12.2	56.8 ± 6.2 ^†¥^	11.2	47.8 ± 4.8	9.9	53.9 ± 4.3	8.0	54.7 ± 5.2	9.66	53.3 ± 10.4	19.6	0.00	0.41
HR_avg_ (%)	56.1 ± 8.5	16.6	71.0 ± 8.5	13.4	71.4 ± 5.8 ^†¥^	8.3	58.2 ± 5.6	9.4	69.2 ± 5.7	8.8	68.4 ± 6.0	9.3	68.1 ± 12.9	19.0	0.00	0.40
HR_max_ (%)	81.6 ± 10.9	15.4	85.2 ± 9.0	12.0	86.5 ± 5.7 ^†¥^	6.7	77.2 ± 8.4	10.8	83.2 ± 6.6	8.3	83.2 ± 6.7	8.3	82.2 ± 15.3	18.6	0.00	0.35
Time in HR Z1 (min.)	5.9 ± 3.4	63.4	1.5 ± 1.4 ^†¥^	110.2	2.1 ± 2.1 ^¥^	136.3	6.3 ± 3.2	59.1	2.5 ± 1.8 ^¥^	110.9	3.2 ± 2.5	127.4	2.4 ± 2.2	93.1	0.00	0.34
Time in HR Z2 (min.)	3.9 ± 2.4	58.0	4.2 ± 2.6 *^†¥^	76.2	6.8 ± 4.1 ^¥^	70.5	3.9 ± 2.2	64.4	6.9 ± 3.3 ^¥^	55.2	8.5 ± 4.4	59.3	25.6 ± 10.9	42.4	0.00	0.74
Time in HR Z3 (min.)	2.1 ± 1.3	58.9	6.2 ± 2.6 *	49.4	8.6 ± 3.8 ^†^	49.4	1.5 ± 1.7	118.9	7.0 ± 3.2	52.4	7.8 ± 3.6	52.1	28.5 ± 10.0	35.1	0.00	0.48
Time in HR Z4 (min.)	1.2 ± 0.9	95.9	4.0 ± 2.8 ^†¥^	88.3	4.2 ± 3.46 ^†¥^	101.2	0.4 ± 0.8	171.3	2.6 ± 2.2	114.0	2.9 ± 2.8	118.7	13.2 ± 10.5	79.4	0.00	0.50
Time in HR Z5 (min.)	0.2 ± 0.3	185.3	0.8 ± 1.03 ^†¥^	150.6	0.7 ± 1.0 ^†^	165.4	0.1 ± 0.1	197.1	0.3 ± 0.4	143.1	0.4 ± 0.5	156.7	2.2 ± 3.6	165.7	0.01	0.29
Distance (m)	515.3 ± 133.0	30.7	708.0 ± 147.0 *^¥^	22.8	842.5 ± 179.3 ^†^	21.7	243.4 ± 84.0	32.7	757.2 ± 127.2	17.3	813.8 ± 192.9	23.9	3034.8 ± 422.4	15.5	0.00	0.56
Max Speed (km·h^−1^)	18.7 ± 4.3	26.0	20.7 ± 3.27	17.7	21.0 ± 2.3	11.7	13.3 ± 5.1	39.8	21.0 ± 2.4	11.9	21.0 ± 2.8	13.9	15.3 ± 3.2	21.2	0.89	0.02
Ave Speed (km·h^−1^)	1.5 ± 0.4	30.9	2.4 ± 0.49 *^¥^	21.8	2.2 ± 0.3 ^¥^	15.0	1.0 ± 0.3	29.0	2.3 ± 0.4 ^¥^	16.0	2.1 ± 0.3	16.1	2.2 ± 0.5	20.5	0.00	0.59
Distance in Speed Z1 (m)	174.6 ± 52.1	33.0	295.0 ± 77.6 *^†¥^	28.0	393.4 ± 97.6 ^†^	25.1	177.9 ± 36.8	21.5	348.0 ± 68.3 ^¥^	20.2	405.7 ± 114.9	27.6	1402.0 ± 233.0	17.9	0.00	0.72
Distance in Speed Z2 (m)	160.1 ± 61.7	40.9	154.6 ± 38.3 *	26.9	171.9 ± 39.7 ^†^	23.8	21.6 ± 25.7	122.2	154.4 ± 28.9	19.9	164.5 ± 42.2	26.8	627.40 ± 96.2	16.9	0.00	0.36
Distance in Speed Z3 (m)	100.9 ± 39.4	49.4	116.9 ± 35.5	33.3	118.4 ± 35.2 ^¥^	31.5	13.9 ± 17.2	139.9	114.0 ± 33.6	30.4	107.3 ± 34.4	33.5	443.9 ± 96.9	24.5	0.04	0.23
Distance in Speed Z4 (m)	30.1 ± 17.9	89.2	68.7 ± 29.6 ^¥^	47.7	71.7 ± 29.8 ^¥^	46.8	6.3 ± 9.7	162.4	65.0 ± 26.6	44.0	57.2 ± 24.4	46.9	255.2 ± 70.7	32.8	0.01	0.31
Distance in Speed Z5 (m)	25.5 ± 13.4	108.2	26.1 ± 18.1	85.9	27.4 ± 16.3	91.1	3.6 ± 5.8	182.3	22.9 ± 13.6	82.1	20.3 ± 14.0	92.5	94.0 ± 37.0	58.3	0.11	0.17
Deceleration Z1 (number)	0.9 ± 0.8	131.1	4.0 ± 2.1 ^¥^	68.9	3.8 ± 2.1 ^¥^	80.8	0.3 ± 0.5	189.9	3.6 ± 1.94 ^¥^	76.4	2.8 ± 1.7	77.3	14.2 ± 11.4	80.3	0.01	0.29
Deceleration Z2 (number)	4.8 ± 2.6	65.6	20.6 ± 6.3	33.0	22.6 ± 7.0	32.3	2.2 ± 2.7	126.3	20.8 ± 5.2	25.9	21.0 ± 5.9	29.4	84.9 ± 14.6	17.2	0.03	0.24
Deceleration Z3 (number)	29.3 ± 9.6	33.4	65.8 ± 16.3 *^†¥^	27.0	83.3 ± 21.5 ^†^	26.2	20.7 ± 10.4	46.2	74.8 ± 13.7 ^¥^	19.1	83.8 ± 23.3	27.9	307.7 ± 42.2	13.7	0.00	0.64
Deceleration Z4 (number)	19.3 ± 7.3	39.6	35.8 ± 9.5 *^†¥^	28.5	43.6 ± 10.2 ^†^	24.5	13.1 ± 5.9	44.2	40.0 ± 8.3 ^¥^	21.9	43.8 ± 11.7	26.8	163.2 ± 27.0	16.5	0.00	0.54
Acceleration Z1 (number)	23.9 ± 8.20	35.0	40.9 ± 10.3 *^†¥^	26.8	53.2 ± 13.1 ^†^	24.9	18.5 ± 6.9	36.2	47.3 ± 10.0 ^¥^	22.2	55.6 ± 15.1	27.2	197.1 ± 38.8	19.7	0.00	0.69
Acceleration Z2 (number)	24.8 ± 9.13	37.8	66.9 ± 17.7 *^†¥^	28.51	81.8 ± 22.9 ^†^	28.6	16.1 ± 10.9	63.6	74.7 ± 15.5 ^¥^	21.2	81.0 ± 23.4	29.1	304.4 ± 43.9	14.4	0.00	0.58
Acceleration Z3 (number)	5.0 ± 2.66	72.0	16.8 ± 4.9 ^¥^	31.7	17.3 ± 5.4 ^¥^	33.2	1.53 ± 1.97	140.9	16.0 ± 4.9	32.1	14.7 ± 4.4	31.0	64.9 ± 12.7	19.6	0.00	0.37
Acceleration Z4 (number)	0.1 ± 0.1	211.08	0.11 ± 0.1	150.9	0.17 ± 0.2	169.3	0.02 ± 0.1	136.9	0.2 ± 0.2	183.4	0.1 ± 0.17	151.0	1.1 ± 1.0	84.0	0.63	0.05
Summa changes (number)	87.9 ± 25.5	29.0	250.9 ± 34.7 *^¥^	13.83	305.8 ± 45.4 ^†^	14.7	87.4 ± 25.2	28.9	277.4 ± 28.4 ^¥^	10.2	302.8 ± 45.3	15.0	1105.3 ± 147.6	14.6	0.00	0.27

Legend: Active time = summa all quarters; HR = heart rate; Min = minimum; Avg = average; Max = maximum; HR zones: Z1 = 50–59%; Z2 = 60–69%; Z3 = 70–79%; Z4 = 80–89%; Z5 = 90–100%; Speed zones: Z1 = 3.00–6.99 km·h^−1^; Z2 = 7.00–10.99 km·h^−1^; Z3 = 11.00–14.99; Z4 = km·h^−1^; 15.00–18.99 km·h^−1^; Z5 ≥ 19.00 km·h^−1^; deceleration/acceleration zones: Z1 = 0.5–1.0 m·s^−2^; Z2 = 1.0–2.0 m·s^−2^; Z3 = 2.0–3.0 m·s^−2^; Z4 ≥ 3.0 m·s^−2^; η_p_^2^ = partial eta-squared (effect size); * = Significantly different from the 2nd quarter; ^†^ = Significantly different from the 3rd quarter; ^¥^ = Significantly different from the 4th quarter.

**Table 3 sensors-24-06900-t003:** Descriptive statistics (mean ± SD) and between-referees variability of external and internal load data of FIBA referees (N = 76 games × 3 ref.).

Variables	Warm-Up		1st Quarter		2nd Quarter		Half Time		3rd Quarter		4th Quarter		Total (Active Time)	
	Mean ± SD	CV%	Mean ± SD	CV%	Mean ± SD	CV%	Mean ± SD	CV%	Mean ± SD	CV%	Mean ± SD	CV%	Mean ± SD	CV%
HR_min_ (beats/min.)	81.4 ± 10.2	12.7	102.1 ± 10.7	10.9	104.9 ± 11.4	11.3	89.1 ± 9.5	10.8	100.1 ± 9.9	10.03	102.4 ± 10.3	10.3	102.4 ± 10.6	10.3
HR_avg_ (beats/min.)	100.5 ± 18.8	27.5	126.8 ± 19.9	23.4	131.4 ± 11.0	8.6	108.4 ± 8.8	8.1	127.8 ± 12.3	10.9	126.6 ± 12.6	11.2	128.2 ± 13.9	10.9
HR_max_ (beats/min.)	144.4 ± 25.2	26.6	151.4 ± 23.2	23.1	159.4 ± 12.0	7.7	143.5 ± 10.1	7.0	153.8 ± 13.7	10.1	153.5 ± 13.8	10.1	154.5 ± 15.7	10.2
HR_min_ (%)	44.2 ± 5.8	13.2	55.4 ± 6.5	12.2	56.8 ± 6.7	12.4	48.2 ± 5.5	11.7	54.2 ± 5.8	10.9	55.4 ± 5.9	10.9	55.4 ± 6.2	11.2
HR_avg_ (%)	54.5 ± 10.7	28.1	68.8 ± 11.62	24.4	71.1 ± 6.7	9.7	58.7 ± 5.3	9.	69.2 ± 7.3	11.8	68.5 ± 7.1	11.6	69.4 ± 8.2	11.8
HR_max_ (%)	78.2 ± 13.5	26.5	82.0 ± 13.2	23.7	86.2 ± 7.0	8.3	77.7 ± 5.9	7.6	83.2 ± 7.7	10.5	83.0 ± 7.6	10.3	83.6 ± 8.9	10.6
Time in HR Z1 (min.)	5.9 ± 3.5	70.1	1.3 ± 1.2	106.5	1.9 ± 2.2	127.3	6.3 ± 3.0	59.6	2.3 ± 2.2	110.2	2.8 ± 3.0	118.2	8.3 ± 8.6	102.6
Time in HR Z2 (min.)	4.3 ± 2.7	72.6	3.7 ± 2.2	76.2	6.5 ± 3.5	70.1	4.1 ± 2.6	72.6	6.7 ± 3.2	55.6	8.5 ± 4.2	56.1	25.4 ± 13.1	51.5
Time in HR Z2 (min.)	2.1 ± 1.3	63.2	6.1 ± 2.9	55.7	8.3 ± 3.9	52.5	1.5 ± 1.1	93.5	7.3 ± 3.2	50.8	8.2 ± 3.9	55.0	29.9 ± 13.9	46.5
Time in HR Z4 (min.)	1.3 ± 0.9	100.5	4.1 ± 2.8	86.8	4.2 ± 3.3	92.3	0.4 ± 0.4	123.9	2.8 ± 2.6	108.6	3.0 ± 2.9	115.6	14.1 ± 11.7	82.7
Time in HR Z5 (min.)	0.2 ± 0.3	159.5	0.9 ± 1.4	158.9	0.7 ± 1.0	158.1	0.1 ± 0.1	139.7	0.3 ± 0.5	166.5	0.4 ± 0.6	156.1	2.3 ± 3.4	149.7
Distance (m)	529.5 ± 210.5	48.3	690.1 ± 131.2	28.3	816.2 ± 127.7	19.2	253.2 ± 43.5	16.3	755.0 ± 93.7	13.9	815.5 ± 101.7	14.0	3076.9 ± 454.3	14.8
Max Speed (km·h^−1^)	18.1 ± 5.9	42.1	20.2 ± 4.6	31.3	21.0 ± 3.4	17.8	13.6 ± 3.7	27.4	21.2 ± 3.3	16.7	20.9 ± 3.2	16.2	20.8 ± 3.6	17.4
Ave Speed (km·h^−1^)	1.6 ± 0.6	48.3	2.3 ± 0.4	28.3	2.2 ± 0.3	16.2	1.0 ± 0.2	17.1	2.3 ± 0.3	13.6	2.1 ± 0.3	13.4	2.3 ± 0.3	14.6
Distance in Speed Z1 (m)	178.2 ± 60.1	44.7	291.2 ± 65.8	31.5	0.5 ± 0.6	151.4	181.4 ± 27.6	15.6	348.0 ± 60.3	18.9	409.8 ± 65.5	17.4	1432.8 ± 264.8	18.5
Distance in Speed Z2 (m)	170.2 ± 97.9	62.3	147.8 ± 36.2	33.5	383.7 ± 73.1	22.2	26.2 ± 18.3	82.5	151.7 ± 27.6	19.7	161.7 ± 27.7	18.9	625.2 ± 124.3	19.9
Distance in Speed Z3 (m)	102.6 ± 77.4	77.7	111.7 ± 33.9	40.3	163.9 ± 32.8	24.7	14.3 ± 11.3	104.2	112.9 ± 27.5	27.3	106.7 ± 29.2	29.9	446.1 ± 120.7	27.1
Distance in Speed Z4 (m)	30.1 ± 26.4	106.5	67.1 ± 30.8	58.6	114.8 ± 30.1	31.4	6.8 ± 5.7	118.7	64.9 ± 29.5	50.5	57.3 ± 27.3	50.2	256.70 ± 119.1	46.4
Distance in Speed Z5 (m)	24.3 ± 28.2	140.8	26.6 ± 24.1	99.1	67.5 ± 31.5	51.9	4.2 ± 5.2	143.8	24.4 ± 20.4	93.7	20.6 ± 18.3	100.8	100.0 ± 90.7	90.6
Deceleration Z1 (number)	0.9 ± 0.9	135.6	3.7 ± 3.1	96.4	265.2 ± 53.9	23.4	0.3 ± 0.4	159.0	3.5 ± 2.9	93.9	2.7 ± 2.3	90.8	13.5 ± 11.3	83.5
Deceleration Z2 (number)	4.8 ± 3.5	78.9	19.9 ± 6.2	40.5	3.7 ± 2.9	92.9	2.2 ± 1.1	80.9	20.5 ± 5.0	26.6	20.9 ± 5.4	29.2	83.0 ± 22.8	27.4
Deceleration Z3 (number)	30.2 ± 10.9	45.5	64.9 ± 14.7	31.1	21.7 ± 6.2	34.7	20.9 ± 4.8	24.3	75.2 ± 12.3	17.8	84.6 ± 13.0	17.4	305.5 ± 55.9	18.3
Deceleration Z4 (number)	19.4 ± 7.3	46.6	35.4 ± 10.4	37.6	80.8 ± 15.9	23.9	13.5 ± 4.4	33.3	39.9 ± 8.9	23.1	44.5 ± 9.2	22.0	162.5 ± 38.6	23.8
Acceleration Z1 (number)	24.4 ± 9.4	47.5	39.9 ± 12.4	39.2	42.6 ± 10.1	26.6	18.9 ± 6.1	32.5	47.1 ± 11.1	25.1	56.1 ± 12.5	23.9	194.3 ± 48.3	24.8
Acceleration Z2 (number)	25.9 ± 10.9	49.9	65.9 ± 16.1	33.1	51.3 ± 12.3	26.2	16.4 ± 5.1	35.9	74.8 ± 13.6	19.4	81.9 ± 15.2	21.2	302.4 ± 61.5	20.3
Acceleration Z3 (number)	4.8 ± 3.2	82.9	16.2 ± 4.8	38.6	79.7 ± 16.6	26.4	1.6 ± 0.9	95.1	15.8 ± 3.9	27.7	14.7 ± 3.7	26.6	63.1 ± 17.3	27.3
Acceleration Z4 (number)	0.1 ± 0.1	165.3	0.1 ± 0.2	161.2	16.4 ± 4.8	34.9	0.1 ± 0.1	163.2	0.2 ± 0.2	163.6	0.2 ± 0.2	163.2	0.4 ± 0.8	188.8
Summa changes (number)	81.4 ± 10.2	12.5	246.2 ± 63.6	25.8	296.3 ± 77.5	26.1	72.8 ± 38.5	52.9	276.9 ± 44.7	16.1	305.4 ± 80.6	26.4	1124.9 ± 64.07	22.8

Legend: Legend: Active time = summa all quarters; HR = heart rate; Min = minimum; Avg = average; Max = maximum; HR zones: Z1 = 50–59%; Z2 = 60–69%; Z3 = 70–79%; Z4 = 80–89%; Z5 = 90–100%; Speed zones: Z1 = 3.00–6.99 km·h^−1^; Z2 = 7.00–10.99 km·h^−1^; Z3 = 11.00–14.99; Z4 = km·h^−1^; 15.00–18.99 km·h^−1^; Z5 ≥ 19.00 km·h^−1^; deceleration/acceleration zones: Z1 = 0.5–1.0 m·s^−2^; Z2 = 1.0–2.0 m·s^−2^; Z3 = 2.0–3.0 m·s^−2^; Z4 ≥ 3.0 m·s^−2^.

## Data Availability

The raw data supporting the conclusions of this article will be made available by the authors upon request.

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
