# Peer review of "Cardiovascular Response and Locomotor Demands of Elite Basketball Referees During International Tournament: A Within- and Between-Referee Analysis"

_sensors, 2024, doi:10.3390/s24216900_

Round 1

Reviewer 1 Report

Comments and Suggestions for Authors

Authors conducted observational and cross-sectional studies within- and between-referee variation (WBRV) in cardiovascular responses (CVR) and locomotor game demands (LMD) in male basketball referees during elite international games,  by examining both preparation (warm-up and re-warm-up) and active game phases. Results indicated higher WBRV, CVR, and LMD during the warm-up phase compared to re-warm-up, with a decrease in CVR and LMD in the second half, suggesting the need for consistent preparation routines and improved half-time reactivation.

1. Authors should describe the intensity and specifics of the training regime, which might influence the outcomes.

2. Each referee engaged between 3 to 9 games, which might introduce variability in the data. Discuss the probability.

3. environmental conditions during the test (e.g., indoor vs. outdoor, temperature) can influence performance. How the authors exclude the possibility? Also, what are the pre-test regimens implemented for the study?

4. Authors can state the rationale for the chosen statistical methods and discussing the potential implications of high variability on the study's findings .

5.Ther might be factors which influences HRavg and HRmax between quarters like  game intensity, referee fatigue.Authors should discuss this possibility to refine the discussion.

Reviewer 2 Report

Comments and Suggestions for Authors

Thank you for the opportunity to review the article titled – Cardiovascular Response and Locomotor Demands of Elite Basketball Referees During International Tournament: A Within and Between-Referee Analysis

Recommendations:

Introduction - to add the hypothesis of the study; to detail the new aspects reported in previous studies on the topic.

Study design - to add the period of the study and recommend; - recommend adding a diagram to highlight the stages of the study.

Conclusions – recommend moving the practical implications to the end of the Discussion section.

Reviewer 3 Report

Comments and Suggestions for Authors

I was given a manuscript to check titled:

 Cardiovascular Response and Locomotor Demands of Elite Bas-2 ketball Referees During International Tournament: A Within- 3 and Between-Referee Analysis.

The primary aim of this study was to assess WBRV of CVR and LMD in male basketball referees during elite, international games in preparation [e.g., warm-up (WU) and re-warm-up (R-WU)] and active game phases. The secondary aim was to explore quarter-by-quarter differences in CVR and LMD. Thirty-five international male referees took part in this study (age, 40.4±5.4 years; body height, 184.9±5.7 cm; body weight 85.1±7.5 kg; BMI, 24.0±1.7 kg×m−2; fat%, 18.8±4.7% and VO2max, 50.4±2.2 L×kg-1×min-1. In total, 64 games (e.g., 192 officiating cases) were analyzed during the FIBA elite men's competition. They officiated 4.5 games on average (range 3-9 games). Each referee used the Polar Team Pro system to measure CVR [e.g., heart rate (HR), time spent in different HR intensity categories] and LMD (e.g., distance covered, maximal and average velocity, number of accelerations). Results showed that the referees had bigger WBRV during the active and preparation (e.g., W-U than R-WU) phase when variables of higher CVR and LMD intensity were observed (e.g., time spent at higher HR zones, distance covered in higher speed zones). The WBRV, CVR, and LMD were higher during WU than R-WU. Moreover, the referees had a lower CVR and LMD in the second half. In conclusion, the referees should establish and follow consistently a game-to-game preparation routine and attempt to spread their on-court preparation time equally within the crew. A half-time preparation routine should be improved to re-establish a sufficient activation level similar to that achieved in pre-game preparation.Although the study has the potentiality of being shared with the scientific community, I believe that the manuscript would benefit from a minor revision with the attempt to better support their experimental setting.

INTRODUCTION: The conceptual framework is limited, the authors should distinctly articulate the empirical evidence that undergirds the proposition they have put forth.

DISCUSSION: The Discussion ought to be enhanced with the prevailing theory. The authors should distinctly delineate the empirical evidence that undergirds their conclusions. Furthermore, they should commence with an initial paragraph elucidating the primary objectives and subsequently the principal outcomes.

CONCLUSION: Conclusions section ought to provide a concise response to the objective of the manuscript. The authors addressed the objective in a limited manner.

I would like to see more of the practical implications. Based on the analyzed variables, how the authors intend to use their findings?

Kind regards
